# Novel Application of Hybrid Anion Exchange Resin for Phosphate Desorption Kinetics in Soils: Minimizing Re-Adsorption of Desorbed Ions

**Zhe Li, Suwei Xu, Ying Li and Yuji Arai ***

Department of Natural Resources and Environmental Sciences, the University of Illinois at Urbana-Champaign, Urbana, IL 61801, USA; zhel2@illinois.edu (Z.L.); suweixu2@illinois.edu (S.X.); yingl@illinois.edu (Y.L.)

**\*** Correspondence: yarai@illinois.edu; Tel.: +1-(217)-244-3602

**Abstract:** The process of phosphate desorption from soils is difficult to measure using stirred batch techniques because of the accumulation of desorbed ions in a bathing solution. To accurately measure the apparent rate coefficient of phosphate desorption from soils, it is necessary to remove the desorbed ions. In this study, a novel hybrid (i.e., iron oxide coated) anion exchange resin was used as a sink to study long-term (seven days) P desorption kinetics in intensively managed agricultural soils in the Midwestern U.S. (total phosphorus (TP): 196–419 mg/kg). The phosphate desorption kinetics in the hybrid anion exchange resin method were compared with those in the other conventional batch desorption method with pure anion exchange resins or without any sink. The extent of P desorption in the hybrid resin methods was >50% of total desorbed phosphate in the other methods. The initial kinetic rate estimated in the pseudo-second-order kinetic model was also highest (3.03–31.35 mg/(g·hr)) in the hybrid resin method when the same soil system was compared. This is because adsorbed P in the hybrid resins was nearly irreversible. The hybrid anion exchange resin might be a new and ideal sink in measuring the P desorption process in soils and sediments.

**Keywords:** desorption; kinetics; phosphate; P; sink; hybrid resin; soil

## 1. Introduction

The desorption process of ions is one of the most important soil chemical processes influencing the availability of inorganic pollutants in soil. Along with the solid-state speciation of contaminants, the rate of release, including desorption and dissolution, is critical in the environmental risk assessment.

The kinetic rate is often evaluated using stirred batch desorption techniques, bathing soil particles in a solution without a sink. However, such methods have a major experimental problem, re-adsorption of desorbed ions to soils (i.e., backward reaction) and or accumulation desorbed ions in a bathing solution. Therefore, the rate does not represent the actual apparent rate coefficient of desorption process in soils. To overcome the problem, the stirred flow method is often used in soil chemistry and geochemistry experiments [1]. This method traps soil particles in a stirred-flow chamber, and the continuous flow of influent is pumped into the chamber, resulting in the continuous removal of desorbed ions from the soil particles. This process will eliminate the accumulation of desorbed ions and minimize the backward reactions. This method works well in fine silt and clay fractions of soils, but it is not suited for natural soils that have sand and coarse silt fractions. In a reaction chamber, soil particles were continuously mixed using a magnetic stir bar. While fine fractions are mixed well in a chamber, coarse fractions (i.e., sand and coarse silt) settle at the bottom of the chamber where the magnetic stir bar is rotating. This will abrade the coatings of sand and silt particles and or break micro-aggregates. In natural soils, coarse particles often have reactive coatings. Therefore, these coarse

fractions can interact with ions like phosphate in the environment [2–4]. Destruction of the natural integrity of soil particles is a major experimental fraud.

To avoid the abrasion of particle coatings and micro aggregates, the overhead mixing method is often suggested. However, a well-homogenized state is difficult to achieve because coarse fractions tend to settle at the bottom a reaction vessel. Variable soil/solution ratios in a water column become an issue when the soil suspension at a constant soil/solution ratio needs to be sampled during the kinetic experiments [5]. Therefore, this overhead mixing method is not ideal for studying the ion desorption in soils.

To accurately measure the desorption rate of natural soils, one must overcome these technical issues in the stirred batch methods and the stirred flow method. If possible, it is ideal to provide a sink to remove desorbed ions from a bathing solution. Anion exchange resins have been widely used to remove phosphate from wastewater through the ion exchange process [6–9]. The matrix of ion exchange resins provides good adsorption sites for desorbed ions, but they predominantly removed desorbed ions via an ion exchange reaction and are not selective toward a specific anion of interest, phosphate [9–11]. The modification of anion exchange resins by coating with other materials can improve their performance as a sink [12–15].

Iron (oxyhydr)oxide is one of the most extensively studied adsorbents to remove anionic pollutants. A Lewis base like phosphate can form inner-sphere complexes on the iron (oxyhydr)oxide surface [16–18]. Thus, their high affinity for anions makes it an efficient adsorbent [19]. However, its small particle size and high reactivity make it difficult to apply for the soil desorption experiments because iron oxide particles cannot be easily separated from bulk soils [15,20–22]. Anion exchange resins coated with iron (oxyhydr)oxide, hybrid resin, are larger and easier to handle and expected to be a promising sink for the removal of phosphate from water and soil solutions [12,15,23]. Resins are more robust materials than filter papers, so they withstand the mixing action more than iron oxide impregnated papers.

Several researchers have shown a high affinity of iron (oxyhydr)oxide loaded resins for anionic pollutants like phosphate, arsenate, and selenate from wastewaters [12,24–26]. However, the use of hybrid anion exchange resin as a sink in soil desorption kinetic experiments has rarely been evaluated in soil science. It was hypothesized that hybrid anion exchange resin is an ideal sink for P to measure the P desorption rate. In this study, a commercially available hybrid anion exchange resin was used in a mesh bag to evaluate phosphate desorption kinetics from agricultural soils in the Midwestern U.S., and the results of kinetic rates were compared with the rates in a respective pure anions exchange resin and in a conventional batch desorption method without a sink. To assure the hybrid resins are an ideal sink for phosphate, the hybrid resin was tested for the maximum phosphate retention capacity and irreversibility. Its performance as a P sink was evaluated in long-term soil desorption experiments.

## 2. Materials and Methods

### 2.1. Materials

The soil samples were collected at the depths of 0–18, 72–90, and 162–180 cm from an intensively managed agricultural land in Douglas County in East–Central Illinois, hereinafter referred to as S_18, S_90, and S_180, respectively. The major soil series at the site is Milford silty clay loam (fine, mixed, superactive, and mesic Typic Endoaquolls). The field received no-till and strip-till practices and has been used to grow corn and soybeans. Soils in this area are poorly drained dark-colored mollisols according to the U.S. Department of Agriculture (USDA) soil taxonomy [27]. All chemicals (Sigma-Aldrich, St. Louis, MO, USA) used in this study are ACS grade unless otherwise mentioned in the text. Ultrapure water (18.2 MΩ·cm) was used to make all solutions.

## 2.2. Anion Exchange Resin and Hybrid Resin

A pure anion exchange resin and an iron oxide-loaded hybrid resin were used as P sinks in the desorption experiments. A pure anion exchange resin AMBERLITE™ HPR 9200 (DuPont Company, Wilmington, NC, USA) was chosen because of its wide application in environmental research. A commercially available product, FerrIX™A33E (Purolite, King of Prussia, PA, USA), was used as a hybrid resin. It shares the same properties of the parent anion exchange resin with AMBERLITE™ HPR 9200. The decision to use the commercial hybrid resin was made to provide easier access to the adsorbent for the scientists who are interested in reproducing similar desorption experiments in different soils.

The total surface area of resins was analyzed using the ethylene glycol monoethyl ether (EGME) method. The resins were first dried in an oven at 40 °C for the hybrid resins and 70 °C for pure resins until they reach constant weight. A temperature of 40 °C was chosen for the hybrid resin so as to not convert the iron oxide coating to hematite. Approximately 0.5 g of dried resin samples were weighed and then placed in Petri dishes. The dish was placed in a vacuum desiccator with another Petri dish filled with a 10 mL EGME solution (Sigma-Aldrich, St. Louis, MO, USA). The evacuation-stabilization-weighting cycle was repeated until the weights were relatively constant. It took about six days. Assuming monolayer coverage, the specific surface area was calculated using the following equation [28]:

$$A = W_g / (W_s \cdot 0.000286) \tag{1}$$

where $A$ = specific surface (m$^2$/g), $W_g$ = weight of EGME retained by the sample after monolayer equilibration (g), and $W_s$ = weight of the dried resin (g); 0.000286 is the mass of EGME required to form a monolayer on 1 m$^2$ of the surface.

Before the soil desorption experiment, the functional group of the pure resin HPR 9200 was saturated with bicarbonate. Twenty grams of resins were shaken in a 1 L polypropylene bottle filled with a 0.5 M NaHCO$_3$ solution for 1 h. This treatment was followed by washing with de-ionized water twice. The resins were air-dried.

In order to effectively separate resins from soils during the desorption experiments, 6.5 × 4.5 cm polyester monofilament mesh bags with 150 μm mesh size were used to enclose the resin beads (Universal Filters, Inc., Asbury Park, NJ, USA). The mesh size is large enough to prevent the diffusion-limited reaction through the bag but small enough not to leach hybrid resin beads. Each bag was filled with 1 g of resin beads and then sewn up to seal the opening. Staples were not used to seal the bag to avoid metal contamination in the system.

## 2.3. Mineralogical Characterization of the Iron Oxide Coating of the Hybrid Resin

The mineralogical analysis of the hybrid resin, FerrIX™A33E, was conducted using an X-ray diffraction method. The finely ground resin powder was placed in the 25-stage sample holder on the instrument, Bruker D-5000 XRD unit (Bruker Corporation, Billerica, MA, USA), and positioned the goniometer to start its angular scan at 45 kV and 30 mA. The lower limit for 2θ was set to 5° and the upper limit to 80°. The scanning rate was set at 2.0°/min. The 2θ was calculated using Bragg's Law (2dsinθ = nλ), and the 2θ-intensity XRD pattern was plotted. Peak assignment and mineral identification were performed using the ICDD database (Powder Diffraction File 4, PDF-4+) with the 2theta position and intensity ratio of each peak.

The Fe mineralogy was also analyzed using Fe K-edge X-ray absorption spectroscopy (XAS) at ID12 at Advanced Photon Source (Argonne, IL, USA). XAS is more sensitive to picking up residual amorphous phases like ferrihydrite. A monochromator consisting of a double-crystal Si (220) at Phi = 0° was used. An incident of X-ray energy was calibrated at the first inflection point (7112 eV) of an Fe foil spectrum and detuned 50% at ~7770 eV. Beam size was 2 mm in width × 1 mm in height. The calibration energy was monitored using an Fe foil during the scan. The transmission measurements were performed in air at room temperature. Spectra were recorded with three regions: 10 eV steps from

6880 to 7090 eV with 1sec. dwell, 0.25 eV steps over the pre-edge from 7090 to 7140 eV with 1 s. dwell, and 0.25 eV steps from k of 1.62 to 14 Å$^{-1}$ with 1sec. dwell. Three spectra were recorded. Acquiring multiple spectra across time allows us to quantitatively evaluate reproducibility. Reference spectra of synthetic ferrihydrite, goethite, hematite, and lepidocrocite were also collected. These minerals were synthesized according to the methods described by Schwertmann and Cornell [29]. All mineral samples were diluted in boron nitride (Sigma-Aldrich, St. Lois, MO, USA) except for the hybrid resin sample. The finely ground hybrid resins were packed in a polycarbonate holder and directly measured. Because of the polymer background, the dilution with BN was not necessary. Spectra were normalized using standard features of the ATHENA software package [30], and a linear combination of XAS reference spectra fit analysis was conducted at a *k* range of 2–11 Å$^{-1}$.

## 2.4. Physicochemical Characterization of Soils

The physicochemical properties of soils were measured using the following standard soil science methods. Soil pH was measured at a soil/water ratio of 1:2 in ultrapure water [31,32]. Organic matter content was measured using a loss-on-ignition method [33]. A hydrometer method was used to determine soil texture [34]. Cation exchange capacity was measured using an ammonium acetate (NH$_4$OAc) method at pH 7 [35]. The concentration of extractable P in soils was measured using the Mehlich 3 method [36] and the Bray I method [37]. The molybdenum blue method with excess ammonium molybdenum [38] was used to determine P concentration in extracted solutions.

## 2.5. Total P, Inorganic P (IP), and Organic P (OP) Fractionation of Soils

Total IP and OP were measured in the soils in duplicates using the sequential extraction method described by Kuo [39]. It uses concentrated sulfuric acid (Thermo Fisher Scientific, Waltham, MA, USA) and dilute NaOH solutions. An approximately 1.0 g air-dried soil sample was mixed with 1.5 mL of concentrated H$_2$SO$_4$ in a 50 mL volumetric flask. After mixing, 2 mL of deionized water was added in 0.5 mL increments while mixing vigorously for 10 s after each addition. Next, 21.5 mL of deionized water was added after cooling to room temperature, and then the sample was filtered through Whatman No.2 filter papers (GE Healthcare, Chicago, IL, USA). The filtrate was saved in a 50 mL centrifuge tube for "acid extracted phosphate" determination using the molybdenum blue method [38]. The soil residue and filter paper were then placed in a 125 mL Erlenmeyer flask. A 49 mL of 0.5 M NaOH solution was added and shaken for 2 h at 80 rpm, and then the soil suspension was filtered with Whatman No.2 filter papers. The filtrate was analyzed for "base extracted phosphate" using the same colorimetric method. The concentration of total IP in the soil was calculated by summing the P concentration in the acid ($P_i^a$) or base ($P_i^b$) extracts.

For total P (TP) determination, a 2 mL solution from the acid or base extract was pipetted into a 50 mL volumetric flask. Both 0.5 g of K$_2$S$_2$O$_8$ (Sigma-Aldrich, St. Louis, MO, USA) and 2 mL of 5.5 M H$_2$SO$_4$ solution were added and digested on a hot plate at 150 °C for 30 min. After cooling down, five drops of *p*-nitrophenol were added. pH was adjusted with 1–10 M NaOH solutions until the color changes to yellow. The P concentration was then determined using the molybdenum blue method [38]. The concentration of TP was calculated by summing the P concentration in acid ($TP^a$) and base ($TP^b$).

Accordingly, the total OP fraction in the initial soil sample was calculated using the following equation [38].

$$P_o = TP^a + TP^b - P_i^a - P_i^b \tag{2}$$

## 2.6. Phosphate Adsorption Isotherm in the Anion Exchange Resin and the Hybrid Resin

To design the desorption kinetic experiments with a sink, it is important not to exceed the maximum phosphate retention capacity of the sink during desorption experiments. To understand the maximum phosphate adsorption capacity of the pure resin and the hybrid resin, P adsorption isotherm experiments in these resins were conducted at 21 ± 0.5 °C.

A 100 mg/L phosphate stock solution was prepared by dissolving disodium phosphate in 10 mM NaCl. Approximately 0.1 g pure resin AMBERLITE™ HPR 9200 or hybrid resin FerrIX™A33E were added into a 50 mL Nalgene high-speed centrifugation tube. Appropriate amounts of the phosphate solution were added to make the initial phosphate concentrations of 3, 10, 20, 30, 40, and 50 mg/L. During the first several hours, pH was manually adjusted with 0.01–0.1M HCl or NaOH. The tubes were mixed on an end-over shaker at 30 rpm for 24 h. The experiments were conducted in duplicate. After 24 h, the resin suspensions were sampled and filtered using a 0.45 µm polyvinylidene fluoride (PVDF) syringe filter. Aliquots were colorimetrically analyzed for the concentration of phosphate [38].

Isotherm data were modeled using Freundlich and Langmuir equations.

For the Freundlich model:

$$q = K_f \cdot C^{\frac{1}{n}} \tag{3}$$

where $q$ is the amount of phosphate adsorbed (mg/g); $C$ is the final equilibrium concentration of phosphate (mg/L); $K_f$ is a parameter related to the adsorption capacity; $n$ is a parameter related to the intensity of adsorption.

For the Langmuir model

$$\frac{C}{q} = \frac{1}{Q_m \cdot K_m} + \frac{C}{Q_m} \tag{4}$$

where $q$ is the amount of phosphate adsorbed (mg/g); $C$ is the final equilibrium concentration of phosphate (mg/L); $Q_m$ represents the maximum adsorption capacity (mg/g); $K_m$ is a parameter related to the bonding strength (L/mg).

To understand phosphate adsorption capacity of the mesh bag during the soil desorption experiment (i.e., background P adsorption), a P adsorption isotherm experiment was repeated using blank mesh bags. The mesh bags were cut into strips and 0.5 g of the strip was put into Nalgene 50 mL centrifugation tubes. Different concentrations (1, 2, 3, 4, and 5 mg/L) of phosphate solutions at pH 7.5 in 10 mM NaCl were added. After shaking the tubes on an orbital shaker at 80 rpm for 24 h, aliquots were colorimetrically analyzed for the P concentration [38].

### 2.7. Irreversibility of Adsorbed Phosphate in Resins

The pure resins or the hybrid resins as a sink should be evaluated for the irreversibility (i.e., desorption of adsorbed phosphate from resins). The irreversibility of adsorbed phosphate was tested in duplicate. Resins were first reacted with phosphate by mixing 0.1 g of resins and 20 mL of a 10 mg/L sodium phosphate solution in 50 mL centrifugation tubes. After shaking the mixture on an orbital shaker at 85 rpm for 24 h, an aliquot was carefully decanted. The mass of wet resins and the entrain solution was recorded. To start the irreversibility test of adsorbed phosphate, a 20 mL P-free solution containing 10 mM NaCl and 5 mM 3-(*N*-morpholino) propanesulfonic acid (MOPS) (Sigma-Aldrich, St. Louis, MO, USA) at pH 7.5 was introduced to the tubes, and the tubes were shaken on an orbital shaker at 85 rpm. After 1, 8, 16 h, 1, 2, 3, 5, and 7 days, the concentration of desorbed phosphate was measured by sacrificing each tube. The P concentration was determined colorimetrically [38].

### 2.8. Phosphate Desorption Kinetics in Soils without P Sink

To evaluate the hybrid resin-based phosphate desorption method, the following three systems are compared: (1) desorption without a sink; (2) desorption using the hybrid resin; (3) desorption using a respective anion exchange resin.

First, the soil phosphate desorption experiment was conducted without a sink. Approximately 5–10 g of air-dried soils was added into 125 mL polypropylene bottles filled with 100 mL of ultra-pure water. In this experiment, pH remained at near soil pH. The bottles were gently shaken on a reciprocal shaker at 80 rpm. As soil suspensions were gently agitated, grinding of coating materials, sample alternation, was not an issue in this desorption method. After 2, 4, 8, 16 h, 1, 2, 3, 5, and 7 days,

suspensions were sampled and filtered through 0.45 μm PVDF syringe filters. The aliquot was colorimetrically analyzed for the P concentrations [38].

*2.9. Phosphate Desorption Kinetics in Soils Using Resin Bags*

A phosphate desorption kinetic experiment was conducted using mesh bags filled with either FerrIX™A33E or AMBERLITE™ HPR 9200 (Figure 1). The resins in mesh bags were first hydrated in a 10 mM NaCl solution and pH was adjusted to the corresponding soil pH with 0.001–0.1 M HCl or NaOH solution. Approximately 5 g of air-dried soil samples were added into 125 mL polypropylene bottles that were filled with 100 mL of ultra-pure water. But ~10 g of soil was used for the soil S_180 because it contains much lower TP than the other two soils (see Section 3.1). A resin-filled mesh bag was placed in each bottle. The bottles were shaken on an orbital shaker at 80 rpm.

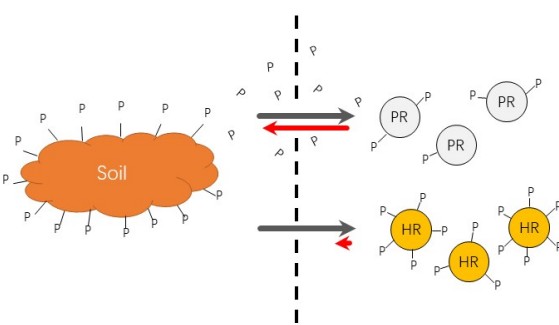

**Figure 1.** A schematic illustration of phosphate (P) desorption systems using pure resins (PR) and hybrid resins (HR) as a P sink. Forward and backward reactions are shown in black and red arrows, respectively. A vertical dotted line indicates a boundary between a mesh bag and a soil solution. Note that more P is retained in HR because of a small backward reaction (i.e., P release) from HR compared to that from PR.

After shaking for 2, 4, 8, 16 h, 1, 2, 3, 5, and 7 days, the resin bag was removed carefully from each bottle and rinsed with 20 mL of de-ionized water once. Then, the resin was dried in a convection oven at 37 °C for 5 h. When resins were recovered from a reaction vessel, the mesh bag containment made an easy separation of resin beads from soil suspensions. Soil particles were not found inside a mesh bag. This is probably because the mesh bags remained in the top layer of the soil solution during shaking. Most soil suspensions remained at the bottom of the reaction vessel during the gentle agitation. To recover adsorbed P from the resins, ~0.5 g of the dried resins was introduced in a 30 mL extractant (0.5 M NaOH + 0.5 M Na$_2$SO$_4$). The chemical composition of this extractant was tested to assure 100% of P recovery under the reaction condition of this experiment. To desorb chemisorbed P in the Fe oxide coating of the hybrid resin, anions with a higher shared charge (e.g., hydroxyl ions) were necessary to extract adsorbed phosphate. The tube was shaken on an orbital shaker at 80 rpm for 24 h. The solution was passed through a 0.45 μm PVDF syringe filter and neutralized with 0.5 M HCl. The P concentration was colorimetrically determined [37]. The experiment was conducted in duplicate.

To understand how the mass of the hybrid resin affects the kinetic rate, the experiments were repeated using 0.5, 1, and 2 g of the hybrid resin in a mesh bag under the same experimental condition. The soil used in this test is S_18.

The data from kinetic experiments were evaluated using pseudo-first-order equation [40], pseudo-second-order equation [41], Elovich equation [42], and intra-particle diffusion model [43].

*Pseudo-first-order equation:*

$$q_t = q_e\left(1 - e^{-k_1 t}\right) \tag{5}$$

which can be expressed in linear form:

$$\ln(q_e - q_t) = \ln(q_e) - k_1 t \tag{6}$$

where $q_t$ is the amount of phosphate adsorbed on unit weight of adsorbents (mg·g$^{-1}$) at time $t$ (h). $q_e$ is the adsorption capacity of adsorbents (mg/g) at equilibrium. The term $k_1$ (g/mg/h) is the first order rate constant. The term $q_e$ and $k_1$ can be calculated by plotting $\ln(q_t - q_e)$ vs. $t$ with linear regression.

　　*Pseudo-second-order equation:*

$$\frac{t}{q_t} = \frac{1}{k_2 q_e{}^2} + \frac{1}{q_e} t \tag{7}$$

where $q_t$ is the amount of phosphate adsorbed on unit weight of adsorbents (mg/g) at time $t$ (h). $q_e$ is the adsorption capacity of adsorbents (mg P/g) at equilibrium. The term $k_2$ (g/mg/h) is the second-order rate constant. The term $q_e$ and $k_2$ can be calculated by plotting $\frac{t}{q_t}$ vs. $t$ with linear regression where the slope and intercept correspond to $\frac{1}{q_e}$ and $\frac{1}{k_2 q_e{}^2}$, respectively.

　　*Elovich equation:*

$$q_t = \frac{\ln(\alpha\beta)}{\beta} + \frac{1}{\beta} \ln(t) \tag{8}$$

where $q_t$ is the amount of phosphate adsorbed on unit weight of adsorbents (mg/g) at time $t$ (h). $\alpha$ (mg/mL/min) is the initial adsorption rate constant, and the parameter b (mL/mg) is related to the extent of surface coverage and activation energy for chemisorptions. The term $\alpha$ and $\beta$ can be calculated by plotting $q_t$ vs. $\ln(t)$ with linear regression where the slope and intercept correspond to $\frac{1}{\beta}$ and $\frac{\ln(\alpha\beta)}{\beta}$, respectively.

　　*Intra-particle diffusion model:*

$$q_t = k_3 t^{1/2} \tag{9}$$

where $q_t$ is the amount of phosphate adsorbed on unit weight of adsorbents (mg/g) at time $t$ (h). $k_3$ is the intra-particle diffusion constant. It can be calculated by plotting $q_t$ vs. $t^{1/2}$ with linear regression where the slope corresponds to $k_3$.

### 2.10. Stability of Fe Oxide Coatings in the Hybrid Resin

To test the stability of Fe coatings of the hybrid resin during desorption experiments, the total iron content of the hybrid resin in a mesh bag was monitored during the same desorption experiment. A mesh bag containing ~1 g of the hybrid resin was added into a 125 mL polypropylene bottle that was filled with a 100 mL 10 mM NaCl solution at pH 7. This experiment was done in triplicate. The bottles were shaken by an orbital shaker at 80 rpm. After seven days, the hybrid resins were removed from the mesh bag and air-dried. Approximately 0.1 g of the hybrid resin was placed in a clean polypropylene bottle to react with a 100 mL extractant solution containing 5% hydroxylamine and 5 M HCl [44,45]. The mixture was placed in an ultrasonic bath (Bransoic, CPX2800) for 5 min and then shaken on an orbital shaker at 80 rpm for 48 h. The mixture was filtered with Whatman No.2 filter papers to screen the pure resin beads, and the total Fe concentration in the filtrate was measured using the spectrophotometric method with 1,10-phenanthroline (Sigma-Aldrich, St. Louis, MO, USA) [46]. The total Fe concentration in these hybrid resins was compared with that of the material before the desorption experiments.

## 3. Results and Discussion

### 3.1. Characterization of Soils and Resins

The physicochemical properties of the soil samples are summarized in Table 1. While OC decreases with increasing depth, IC increases at the lower depth. This is due to the presence of carbonates in subsoils [47]. Soil pH increases with increasing depth from 18 to 180 cm. The weakly acidic pKa of carboxylic acids in organic matter buffer at near-neutral at the surface. Slightly alkaline pH in subsoils at the depth of 90–180 cm is controlled by calcite and dolomitic materials [47]. Because of near-neutral to slightly alkaline pH, CEC is high (23–29 cmol$_c$/kg) throughout the profile. %base saturation is ~96% in all samples.

**Table 1.** Selected physicochemical properties and P concentrations of Douglas County soils.

| Soil Sample ID | Depth | pH | %IC [†] | %OC [†] | CEC [†] | %base [†] | M3P [†] | B1P [†] | OM [†] | Texture | TIP [†] | TOP [†] | TP [†] |
|---|---|---|---|---|---|---|---|---|---|---|---|---|---|
| | cm | | ——% by wt.—— | | cmol$_c$/kg | % | ---mg/kg--- | | % | | ————mg/kg———— | | |
| S_18 | 0–18 | 6.95 | 0.11 (±0.03) [‡] | 2.25 (±0.01) | 26.72 (±2.86) | 95.69 (±0.01) | 14.50 (±0.71) | 10.00 (±1.41) | 4.27 (±0.02) | SC [§] | 251.90 (±0.18) | 139.79 (±17.98) | 391.69 (±18.16) |
| S_90 | 72–90 | 8.03 | 0.27 (±0.06) | 0.80 (±0.01) | 23.23 (±0.46) | 96.05 (±0.31) | 5.00 (±0.00) | 2.00 (±0.00) | 2.12 (±0.02) | SCL [§] | 363.74 (±11.38) | 54.77 (±16.30) | 418.51 (±4.91) |
| S_180 | 162–180 | 8.20 | 2.18 (±0.11) | 0.54 (±0.02) | 28.80 (±1.19) | 96.37 (±0.03) | 1.00 (±0.00) | <1 | 1.39 (±0.01) | SC | 189.35 (±7.17) | 7.05 (±1.11) | 196.41 (±8.28) |

[†] %IC = inorganic carbon content; %OC = organic carbon content; CEC = cation exchange capacity; %base = percent base saturation; M3P = Mehlich III extractable phosphorus; B1P = Bray I phosphorus; OM = organic matter content; TIP = total inorganic phosphorus; TOP = total organic phosphorus; TP = total phosphorus. [‡] Values in the parentheses are the standard deviations. [§] SC = silty clay; SCL = silty clay loam.

In soil S_18, the results of agronomic soil P tests are 10–14.5 mg/kg, which are lower than the recommended level for corn, according to the Illinois Agronomy Handbook [48]. However, the total P in the soils is high (~390 mg/kg) at the surface soil. The majority of P is in the inorganic form of P, and the OP fraction is ~140 mg/kg in S_18. The content of OP decreases from 140 to 7 mg/kg with increasing depth from 18 to 180 cm. However, the IP fraction remains high (~360 mg/kg) up to 90 cm. The S_180 sample contains ~190 mg/kg of IP.

The properties of resins used in this study are shown in Table 2. Both the pure resin AMBERLITE™ HPR9200 and the hybrid resin FerrIX™A33E have polystyrene-DVB as their polymeric matrix. The structure and resin functional groups are the same. The particle size of the pure resin and the hybrid resin is ~640 and ~750 μm. XRD analysis indicates that the mineralogy of iron oxides in the hybrid resin is predominantly goethite (Figure 2a). The LC fitting of XAS reference spectra analysis also shows the same results (Figure 2b). Oscillations and spectra features are nearly identical to those of goethite (Figure 2b). Ferrihydrite was not detected in the XAS analysis.

**Table 2.** General characteristics of the pure anion exchange resin and the hybrid resin used in this study.

| Resin | Matrix | Structure Type | Functional Groups | Physical Form | Particle Size (μm) | Iron Content (mg/g) | Specific Surface Areas (m$^2$/g) |
|---|---|---|---|---|---|---|---|
| AMBERLITE™ HPR9200, pure anion exchange resin | Polystyrene-DVB | Macroporous | Strong-base | White, spherical beads | 640 ± 50 | 0 | 1625.1 ± 20.9 |
| FerrIX™A33E, hybrid resin | Polystyrene-DVB | Macroporous | Strong-base | Brown, spherical beads | 750 ± 150 | 196.0 ± 3.0 | 810.4 ± 38.2 |

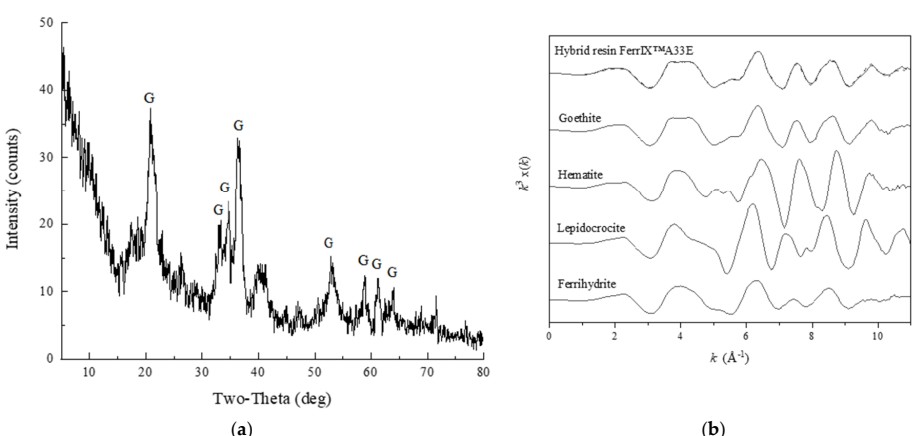

(a)      (b)

**Figure 2.** (**a**) XRD pattern of the ground hybrid resin FerrIX™A33E. Identified peaks correspond to goethite (G), and (**b**) linear combination fit of reference compound fit of a normalized k$^3$ weighted spectrum of the hybrid resin with spectra of reference minerals. Dotted line indicates fit. The fit results show the Fe oxide mineralogy of the hybrid resin is 100% goethite with r-factor of 0.01.

The total surface area of FerrIX™A33E (i.e., hybrid HPR9200), 810.4 ± 38.2 m$^2$/g, is significantly lower than that of HPR9200 (1625.1 ± 20.9 m$^2$/g), which could be explained by the surface coverage of small goethite particles over the porous structure of pure anion exchange resin.

The stability of the Fe oxide coating on FerrIX™A33E was tested in a control system without soils. At $t = 0$, the iron content of the hybrid resin is 195 ± 3 mg Fe/g (Table 2). After shaking a mesh bag containing FerrIX™A33E in a 10 mM NaCl solution for seven days, there was a negligible change in total Fe in FerrIX™A33E. The iron content was 194 ± 2 mg Fe/g. This suggests that the Fe oxide coating of the hybrid resin remains stable during the desorption experiment. The Fe coatings did not leach outside of the mesh bag during the desorption experiments.

## 3.2. Phosphate Adsorption Isotherm in the Resins

The results of the adsorption isotherm experiment show a typical L-shape curve for both resins, indicating that the adsorbates have high affinity at low surface coverage (Figure 3a). When the initial P concentration was at 10 mg/L, ~76% of P was adsorbed by the pure resin, while ~98% was adsorbed by the hybrid resin. Interestingly, the hybrid resin showed a greater affinity for P at the $C_{eq}$: 0–15 mg/L, and the pure resin had a greater affinity for P at $C_{eq} > 15$ mg/L. It is important to note that the adsorption isotherm data in Figure 3a were corrected for the residual P adsorption in mesh bags. Phosphate adsorption by a blank polyester mesh bag was <0.04 mg/g (Figure 3b). The P adsorption capacity of the hybrid resin at near-neutral pH is within the same order of magnitude with that of other adsorbents (e.g., iron oxyhydroxides, zerovalent iron, layered double hydroxides, Zr- and La-containing hydroxides) [49]. However, the hybrid resin is more suitable for the proposed mesh bag-based desorption method than the other adsorbents because its large particle size makes it easy to trap in a mesh bag and to recover or separate from soil solutions.

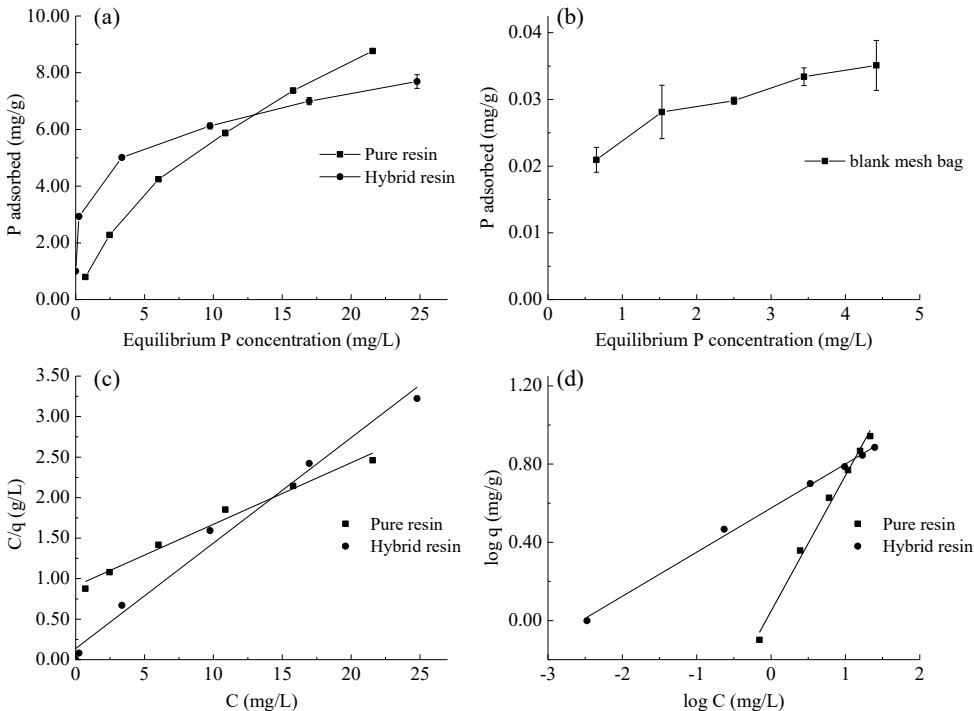

**Figure 3.** (**a**) Phosphate adsorption isotherm at pH 7.5 in the pure anion exchange resin and the hybrid resin; (**b**) phosphate adsorption isotherm at pH 7.5 in a blank mesh bag; (**c**) Langmuir model of the isotherm data in Figure 3a; (**d**) Freundlich model of the isotherm data in Figure 3b.

To compare the affinity of these adsorbents, the adsorption isotherm data of the pure resin were modeled using Freundlich and Langmuir equations. The results are summarized in Table 3.

The model fit in both resins shows an excellent fit. Although these models do not describe the adsorption mechanisms, model parameters can be used to compare the P adsorption capacity of different adsorbents. In the Langmuir model, the maximum adsorption (mg P/g), $Q_m$, indicates that the pure resin has greater adsorption than the hybrid resin. This was observed in Figure 3a. However, in the Freundlich model, the adsorption capacity ($K_f$) of the hybrid resin is much greater than that in the pure resin. Each adsorbent seems to have different properties for P.

**Table 3.** Parameters of isotherm models in phosphate adsorption by resins. $Q_m$ and $K_m$ represent the maximum adsorption (mg P/g) and a parameter related to the bonding strength (L/mg) in the Langmuir model, respectively. $K_f$ and *n* are a parameter related to the adsorption capacity and a parameter related to the intensity of adsorption in the Freundlich model, respectively.

| Resin | Langmuir Model | | | Freundlich Model | | |
|---|---|---|---|---|---|---|
| | $Q_m$ (mg/g) | $K_m$ (L/g) | $R^2$ | *n* | $K_f$ | $R^2$ |
| AMBERLITE™ HPR9200 (Pure resin) | 13.16 | 1.10 | 0.983 | 1.44 | 1.12 | 0.993 |
| FerrIX™A33E (Hybrid HPR9200) | 7.69 | 7.29 | 0.989 | 4.43 | 3.76 | 0.997 |

Using the isotherm data of these resins, it was assured that P adsorption capacity of the two resins is more than enough to be a P sink during the desorption experiments, even 100% of TP (TP < 0.5 mg P/g) is released from the soils. The ideal soil:resin ratio was determined to be 5–10 g of soils:1 g of resin.

### 3.3. Phosphate Irreversibility Tests in the Resins

To evaluate the potential release of adsorbed P in the resins, the desorption of P from P adsorbed resins was tested. The result of the irreversibility test shows a major difference between the pure resin and the hybrid resin serving as P sinks (Figure 4). After providing the total P loading level of 2 mg/g, the resins were shaken in a P-free solution for up to seven days. The cumulative percentage of desorbed P is shown in Figure 4. Approximately, 13.3% of the adsorbed P in the pure resin was released into solution after seven days, and 12.9% of the P had already desorbed within 1 h. It is clear that the pure anions exchange resin is a good sink for the P desorption experiment. In contrast, less than 0.3% of adsorbed P was desorbed from the hybrid resin, which should be accounted for by the inner-sphere complexation of phosphate on the iron oxide coating in the hybrid resins [16,17,50]. The irreversibility test indicates that the hybrid resin had a near-negligible backward reaction (i.e., desorption). The hybrid resin is a better P sink in soil desorption studies.

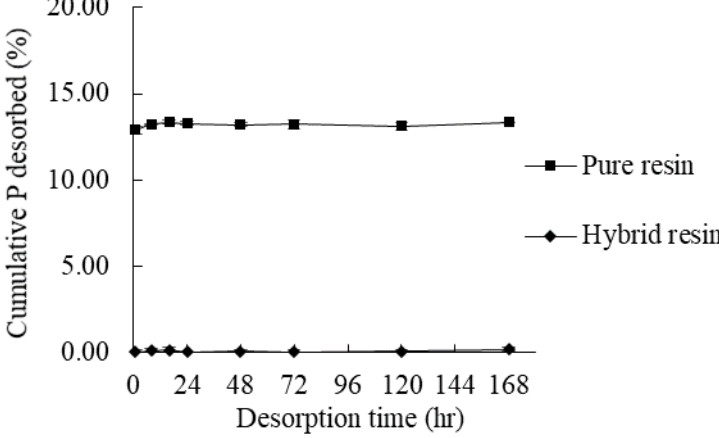

**Figure 4.** Phosphate release from a phosphate adsorbed pure resin and a phosphate adsorbed hybrid resin up to seven days. The initial phosphate loading level in both resins was 2 mg/g.

## 3.4. Phosphate Desorption in Soils

3.4.1. The Extent of P Desorption from Soils

　　The result of P desorption is shown in Figure 5 and Table 4. The amount of P desorbed varies among the three soil samples. With or without a P sink, the extent of P desorption after 168 h (seven days) follows the order of S_18 > S_90 > S_180. This was not surprising because the soils at 0–90 cm contained more TP than subsoils (Table 1). The extent of P desorption was vastly different among the different desorption methods. For each soil, the largest P desorption after seven days is always found in the hybrid resin system, followed by the pure resin system and the system without a P sink. It is important to note that under the fixed mass of adsorbents, the total surface area of pure anion exchange resin, 1625.1 ± 20.9 m$^2$/g, is two times greater than that of the hybrid resin, FerrIX™A33E (i.e., 810.4 ± 38.2 m$^2$/g). This suggests that a difference in surface area is not the factor in removing P from the bathing solution in this experiment.

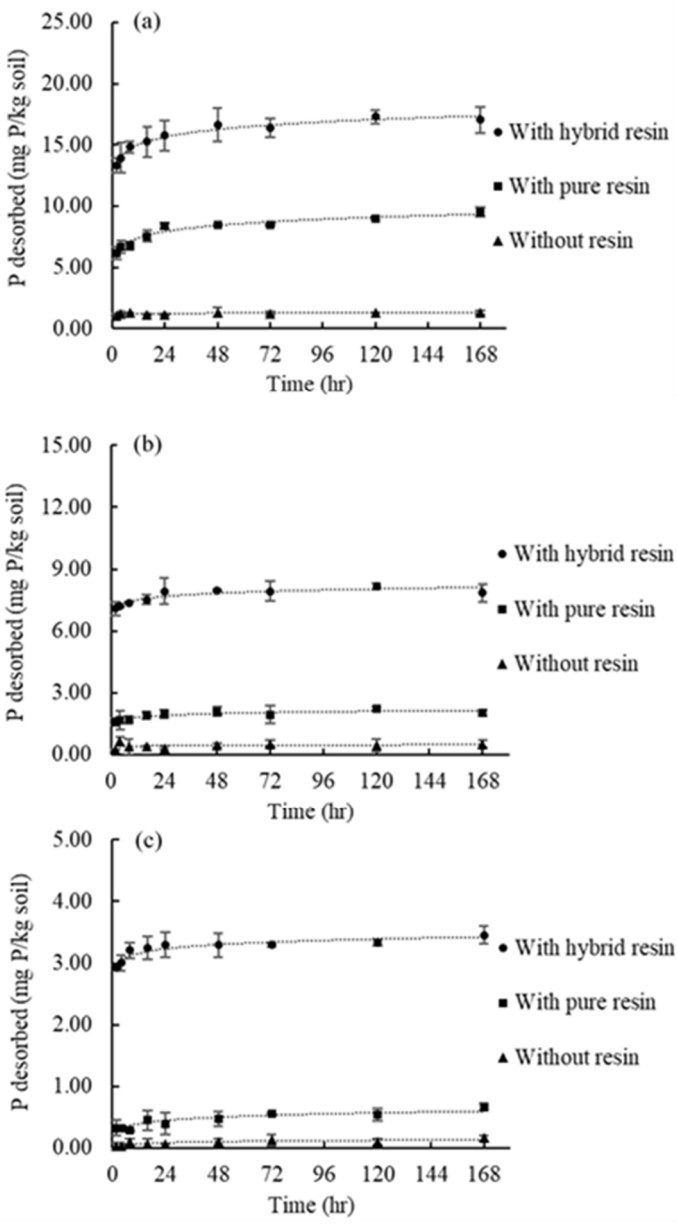

**Figure 5.** Phosphate desorption kinetics from soils (**a**) S_18, (**b**) S_90, and (**c**) S_180 in different desorption systems.

**Table 4.** Phosphate desorption from P rich soils in different desorption systems. The total desorbed P is described in mg/kg, and the values in parentheses represent the percentage of total desorbed P with respect to total P in each soil.

| Soil Sample ID | Desorption Systems | Time (h) | | | | | | | | |
|---|---|---|---|---|---|---|---|---|---|---|
| | | 2 | 4 | 8 | 16 | 24 | 48 | 72 | 120 | 168 |
| S_18 | 1 g of hybrid resin | 13.34 (3.4) | 13.95 (3.6) | 14.82 (3.8) | 15.25 (3.9) | 15.77 (4) | 16.64 (4.2) | 16.38 (4.2) | 17.29 (4.4) | 17.03 (4.3) |
| | 1 g of pure resin | 6.15 (1.6) | 6.67 (1.7) | 6.75 (1.7) | 7.56 (1.9) | 8.37 (2.1) | 8.46 (2.2) | 8.46 (2.2) | 8.98 (2.3) | 9.5 (2.4) |
| | No resin | 1.06 (0.3) | 1.25 (0.3) | 1.32 (0.3) | 1.12 (0.3) | 1.12 (0.3) | 1.32 (0.3) | 1.25 (0.3) | 1.32 (0.3) | 1.32 (0.3) |
| | 0.5 g hybrid resin | 12.38 (3.2) | 12.87 (3.3) | 13.97 (3.6) | 14.79 (3.8) | 15.44 (3.9) | 15.97 (4.1) | 16.46 (4.2) | 16.71 (4.3) | 16.87 (4.3) |
| | 2 g of hybrid resin | 13.40 (3.4) | 14.26 (3.6) | 14.95 (3.8) | 15.65 (4.0) | 15.97 (4.1) | 16.67 (4.3) | 16.99 (4.3) | 17.12 (4.4) | 17.32 (4.4) |
| S_90 | 1 g of hybrid resin | 7.09 (1.7) | 7.19 (1.7) | 7.37 (1.8) | 7.53 (1.8) | 7.94 (1.9) | 7.95 (1.9) | 7.93 (1.9) | 8.15 (1.9) | 7.85 (1.9) |
| | 1 g of pure resin | 1.6 (0.4) | 1.66 (0.4) | 1.69 (0.4) | 1.91 (0.5) | 1.99 (0.5) | 2.1 (0.5) | 1.96 (0.5) | 2.24 (0.5) | 2.03 (0.5) |
| | No resin | 0.26 (0.1) | 0.66 (0.2) | 0.4 (0.1) | 0.4 (0.1) | 0.33 (0.1) | 0.46 (0.1) | 0.53 (0.1) | 0.46 (0.1) | 0.53 (0.1) |
| S_180 | 1 g of hybrid resin | 2.93 (1.5) | 3 (1.5) | 3.21 (1.6) | 3.24 (1.7) | 3.29 (1.7) | 3.29 (1.7) | 3.29 (1.7) | 3.33 (1.7) | 3.45 (1.8) |
| | 1 g of pure resin | 0.33 (0.2) | 0.32 (0.2) | 0.29 (0.1) | 0.46 (0.2) | 0.4 (0.2) | 0.48 (0.2) | 0.56 (0.3) | 0.54 (0.3) | 0.67 (0.3) |
| | No resin | 0.03 (0) | 0.03 (0) | 0.1 (0) | 0.07 (0) | 0.07 (0) | 0.1 (0) | 0.13 (0.1) | 0.1 (0) | 0.16 (0.1) |

To be specific, the topsoil S_18 released 1.32 mg P/g soil without sink after seven days, which was 0.34% of its total P (Figure 5a and Table 1). In the presence of the pure anion exchange resin, the extent of P desorption increased to 9.50 mg P/g (~2.43% of TP). In the case of the hybrid resin, it was even greater. Approximately 17 mg/kg (~4.35% of TP) was desorbed after seven days. The same trend was also observed in the other two subsoils. The soil S_90 released only 0.53 mg/kg (~0.13% of TP) without any sink, but the total desorbed P increased to 7.90 mg P/g (~1.88% of TP) with the hybrid resin as a sink. The pure resin was not as effective as the hybrid resin. In the case of soil S_180, which contains the lowest TP, the efficiency in removing desorbed P was best in the hybrid resin. Only the hybrid resin was able to desorb appreciable amounts of P (~3.45 mg P/g), but no more than 0.67 mg P/g (~0.3% of TP) was desorbed with the pure resin or without any sink.

In summary, the hybrid resin performed best in soils that have a wide range of total desorbable P, the extent of P desorption. Unfortunately, the traditional batch method without any sink or the use of pure anion exchange resin did not produce good results. The backward reaction or accumulation of desorption ions is likely the reason. In the batch system, re-adsorption of desorbed phosphate was readily occurring. In the anion exchange pure resin system, the sink did not behave like a finite sink. As reported in the irreversibility test section, adsorbed P in the pure resin seems to desorb back into the bathing solution. To further validate the hybrid resin-based desorption method, P desorption kinetics in three desorption methods were evaluated using several different models.

### 3.4.2. Phosphate Desorption Kinetics

As shown in Figure 5, the soil P desorption in all desorption methods shows fast desorption within 24 h and then gradually reaches a steady-state in the next several days. The P desorption kinetics were evaluated using different kinetic models. They are the pseudo-first-order model [40], pseudo-second-order model [41], Elovich model [42], and intra-particle diffusion model [43]. Overall, the pseudo-second-order model fits best for the P desorption data in different desorption methods (Table 5). Figure 6 shows the results of the linear fitting of these models to the data in each case. Based

on the goodness of it, the pseudo-second-order model fit best and other models did not fit the data well. The parameters of these kinetic models for the P desorption process are discussed below.

**Table 5.** Parameters of kinetic models in the phosphate desorption data shown in Table 4.

| Soil Sample ID | Desorption Systems | Pseudo-First-Order Model | | | Pseudo-Second-Order Model | | | | Elovich Model | | | Intra-Particle Diffusion Model | | |
|---|---|---|---|---|---|---|---|---|---|---|---|---|---|---|
| | | $k_1$ (1/h) | $q_e$ (mg/g) | $R^2$ | $k_2$ (g/(mg h)) | $q_e$ (mg/g) | $k_2 q_e^2$ (mg/(g h)) | $R^2$ | $\alpha$ (mg/(L h)) | $\beta$ (L/mg) | $R^2$ | $k_d$ (mg/(g h$^{1/2}$)) | $C$ | $R^2$ |
| | 1 g of hybrid resin | 0.031 | 3.014 | 0.482 | 0.037 | 17.241 | 11.099 | 1.000 | $1.72 \times 10^6$ | 1.128 | 0.972 | 0.311 | 13.726 | 0.844 |
| | 1 g of pure resin | 0.038 | 4.734 | 0.763 | 0.029 | 9.461 | 2.601 | 0.998 | $1.45 \times 10^3$ | 1.360 | 0.957 | 0.266 | 6.269 | 0.882 |
| S_18 | No resin | 0.026 | 0.059 | 0.342 | 0.466 | 1.328 | 0.821 | 1.000 | $5.93 \times 10^{10}$ | 25.316 | 0.338 | 0.015 | 1.141 | 0.335 |
| | 0.5 g resin | 0.043 | 5.362 | 0.905 | 0.031 | 17.007 | 8.873 | 1.000 | $5.59 \times 10^4$ | 0.929 | 0.982 | 0.376 | 12.778 | 0.843 |
| | 2.0 g resin | 0.041 | 4.278 | 0.883 | 0.039 | 17.391 | 11.848 | 1.000 | $2.08 \times 10^6$ | 1.125 | 0.982 | 0.309 | 13.946 | 0.834 |
| | 1 g of hybrid resin | 0.021 | 0.659 | 0.326 | 0.495 | 7.962 | 31.348 | 1.000 | $3.73 \times 10^{12}$ | 4.376 | 0.845 | 0.076 | 7.206 | 0.660 |
| S_90 | 1 g of pure resin | 0.019 | 0.461 | 0.316 | 0.443 | 2.097 | 1.949 | 0.996 | $1.74 \times 10^4$ | 7.837 | 0.821 | 0.044 | 1.644 | 0.675 |
| | No resin | 0.031 | 0.272 | 0.581 | 0.429 | 0.520 | 0.116 | 0.988 | $3.01 \times 10^5$ | 43.668 | 0.089 | 0.009 | 0.391 | 0.103 |
| | 1 g of hybrid resin | 0.027 | 0.502 | 0.729 | 0.258 | 3.423 | 3.029 | 0.999 | $5.08 \times 10^{11}$ | 10.040 | 0.879 | 0.034 | 3.020 | 0.727 |
| S_180 | 1 g of pure resin | 0.028 | 0.506 | 0.753 | 0.158 | 0.651 | 0.067 | 0.979 | 1.201 | 13.158 | 0.831 | 0.030 | 0.268 | 0.889 |
| | No resin | 0.022 | 0.152 | 0.700 | 0.367 | 0.151 | 0.008 | 0.882 | 0.040 | 41.322 | 0.724 | 0.009 | 0.032 | 0.733 |

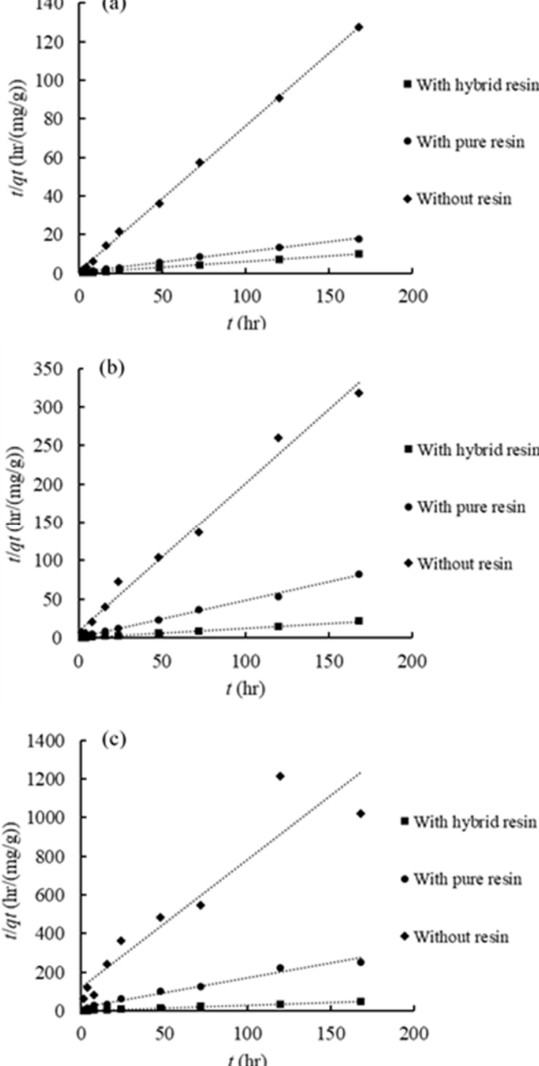

**Figure 6.** Linear plot of pseudo-second-order model for the phosphate desorption data shown in Figure 5. (**a**) S_18, (**b**) S_90, (**c**) S_180.

(1) Pseudo-First-Order Model and Pseudo-Second-Order Model

For three soils, the pseudo-first-order model does not fit well with the kinetic data ($R^2 < 0.8$) (Table 5). There is no clear trend in the rate constant among different desorption conditions or different soil samples. However, the pseudo-second-order model fit resulted in very high $R^2$ values for the kinetic data of all soils, especially for the batch with hybrid resins ($R^2 > 0.999$). The hybrid resin systems also have a larger rate constant ($k_2$) and the adsorption capacity of adsorbents at equilibrium ($q_e$) than the pure resin systems. The batch without any sink had the smallest $q_e$ among three systems, but the rate constant did not show this trend. It should be noted that the $R^2$ of these models becomes low in the system without resins because of a large standard deviation within the small signal (see Figure 5c). The term $k_2 q_e{}^2$ from the pseudo-second-order model can be used as an indication of the initial rate of a desorption process [51]. As shown in Table 5, the initial rate of P desorption follows the trend of hybrid resin > pure resin > no resin. However, among the three soil samples, S_90 has a higher initial P desorption rate than the topsoil S_18, which could be due to the different P speciation in the soils. Inorganic P made up a much higher composition of total P in S_90 than in S_18. It is the fact that S_90 desorbed P faster at the beginning but slower at the end of experiments.

(2) Elovich Model and Intra-Particle Diffusion Model

Judging from the $R^2$ values, the Elovich model and the intra-particle diffusion model have a good fit for the desorption data when the resins are present as a P sink. The goodness of fit in the Elovich model ($0.82 < R^2 < 0.98$) is greater than that of the pseudo-first-order model, but is not as good as that of the pseudo-second-order model. The Elovich model does not fit well to describe the data of the system without resins. The $R^2$ of the intra-particle model is slightly lower than that of the Elovich model. The intra-particle model works better in the resin systems than in the system without resins. The diffusion rate constant was always highest when the hybrid resin is present as P sink for the three soils, possibly suggesting the diffusion-limited P exchange process in the hybrid resins.

In summary, soils could desorb more P when anion exchange resins are present in a reaction vessel. However, the hybrid resins work better to remove P from bathing solutions (i.e., the extent of P desorption) than the pure resins do due to minimized backward reaction during the desorption experiment. In terms of kinetic model fit, the pseudo-second-order model was the best, and the parameter, $k_2 q_e{}^2$, can be used to describe the initial rate of a desorption process.

### 3.4.3. Phosphate Desorption Kinetic Rate Affected by the Mass of Resin

The desorption experiments were repeated in the soil S-18 using different amounts of the hybrid resin to evaluate the effects of resin mass on the kinetic rate. The results are summarized in Figure 7 and Table 5. The biphasic desorption kinetic process was observed in three systems. A fast P release was followed by a slow desorption process. The extent of total desorbed P is similar (i.e., within error bars of each data point at the same sampling time) in these soils. After 168 h, ~16.3 mg/kg of P was released in all systems.

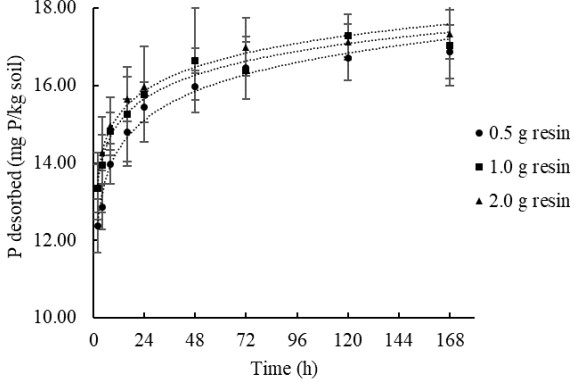

**Figure 7.** Phosphate desorption kinetics in soil S_18 as a function of the hybrid resin mass (0.5–2 g).

Figure 8 shows the linear plots of the pseudo-first-order model, the pseudo-second-order model, the Elovich model, and the intra-particle diffusion model for the P desorption kinetic data shown in Figure 7. The fitting parameters associated with these models are summarized in Table 5 with the $R^2$ values. As previously discussed, the pseudo-second-order model provides the best fit in modeling the P desorption kinetic data (i.e., $R^2 = 1$).

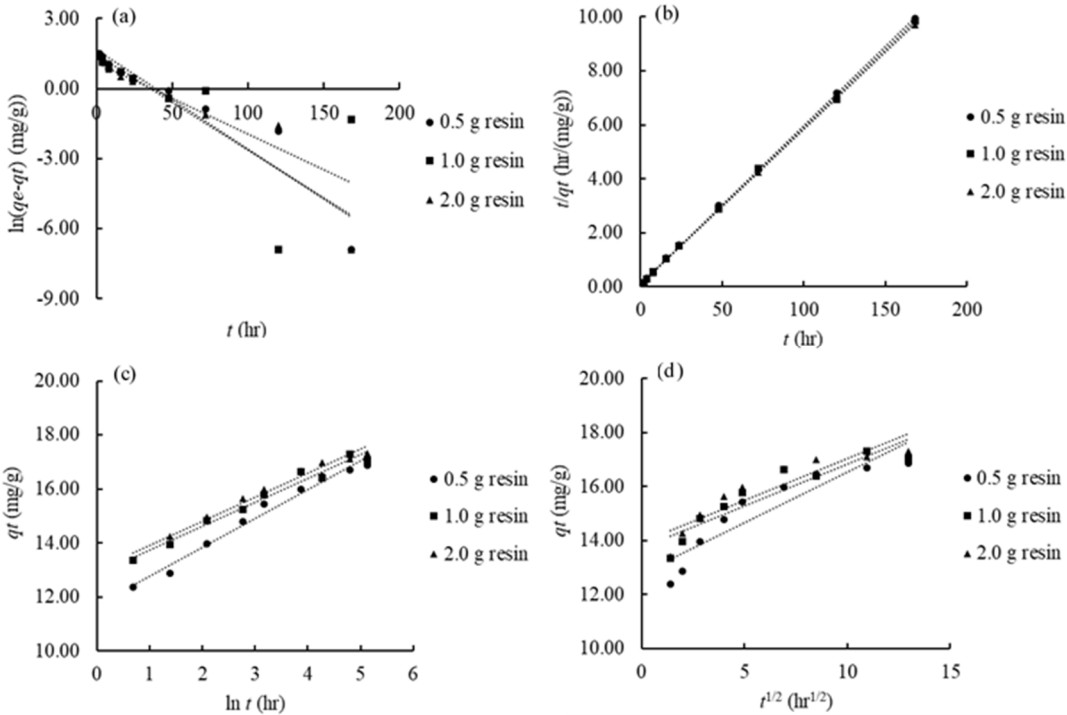

**Figure 8.** Linear plot of (**a**) pseudo-first-order model, (**b**) pseudo-second-order model, (**c**) Elovich model, (**d**) intra-particle diffusion model for the P desorption kinetic data shown in Figure 8.

To assess how the mass of hybrid resin affects the kinetic rate, the same model was used to estimate the initial kinetic rate, $k_2 q_e^2$ from the pseudo-second-order model. The rate increases from ~8.8 to 11.8 mg/g·h when increasing the mass of hybrid resin from 0.5 to 2 g. This suggests that the experimental design and the mass of the resin in the desorption system, can influence the rate of P release and is critical in evaluating the kinetic process.

## 4. Conclusions

This study shows one of the best batch P desorption methods available in the field. The use of the hybrid anion exchange resins as a P sink was very efficient in removing desorbed P from the agricultural soils. Overall, the iron oxide-coated hybrid resins were a better P sink than the respective pure anion exchange resins. The results of the hybrid resin system showed that the extent of P desorption and the initial kinetic rate, $k_2 q_e^2$, were greater than those in the conventional batch methods with and without pure anion exchange resins. This is because adsorbed P in the hybrid resin was nearly irreversible whereas the pure anions exchange resin released ~13% of adsorbed P. This indicates that the backward reaction (i.e., re-adsorption) in the hybrid resin system was minimized. The pseudo-second-order model was successfully used in evaluating the P desorption kinetic data in the hybrid resin.

There are several key points when the hybrid resin-based desorption method is adapted. The maximum adsorption capacity of the hybrid resin sink should be tested to assess a proper soil/resin ratio in the experiment. pH should be maintained or adjusted at the relevant soil pH values during the experiments. The irreversibility test of P from P adsorbed hybrid resins should also be tested under the specific reaction conditions. Minimizing the backward reaction (i.e., re-adsorption) is

a key to succeed the desorption kinetic experiments using a hybrid resin as a sink. When evaluating the kinetic rate of multiple soil samples, the amount of resin in a mesh bag should be consistent since it could influence the initial kinetic rate.

The implication of this study includes that the hybrid anion exchange resin method can be applied to the soil desorption kinetic study of other anions like arsenate in coarse-textured soils and sediments. It is highly recommended to assess the P desorption behavior in sandy soils and vadose zone sediments that cannot be stirred in a reactor to retain the natural integrity of mineral coatings and aggregates. One could also consider the use of different hybrid ion exchange resins. The different types of hybrid ion exchange resins should be further evaluated for the assessment of desorption behavior of other inorganic contaminants (e.g., oxyanions, metal(loid)s, and oxocations) in soils and sediments.

**Author Contributions:** Z.L. and Y.A. drafted the manuscript. Z.L. conducted all macroscopic experiments. S.X. collected and characterized soil samples. Y.L. synthesized the minerals, collected, and analyzed the XAS data. Z.L. handled the XRD analysis. Y.A. was the Principal Investigator for the project, supervisor for Z.L., S.X., and Y.L. and provided rewrites and revisions of the manuscript. Authors have read and agreed to the published version of the manuscript. All authors have read and agreed to the published version of the manuscript.

**Funding:** This research was funded by the United States Department of Agriculture (# 2016-67019-25268), Illinois Nutrient Research and Education Council (#2016-4-360347-203), and China Scholarships Council (# 201706300011) for supporting this project financially. Use of the Advanced Photon Source was supported by the U. S. Department of Energy, Office of Science, Office of Basic Energy Sciences, under Contract No. DE-AC02-06CH11357.

**Acknowledgments:** We would like to thank a cooperating agronomist, Lowell Gentry, who allowed us to sample soils from intensively managed agricultural fields in Illinois and provided management history of corn-soybean fields.

**Conflicts of Interest:** The authors declare no conflict of interest; the funders had no role in the design of the study; in the collection, analyses, or interpretation of data; in the writing of the manuscript; nor in the decision to publish the results.

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
