# Peer review of "Novel Application of Hybrid Anion Exchange Resin for Phosphate Desorption Kinetics in Soils: Minimizing Re-Adsorption of Desorbed Ions"

_soilsystems, doi:10.3390/soilsystems4020036_

Round 1

Reviewer 1 Report

A well written paper on an important topic. The study contains valuable data, satisfactory selection of methodologies and strategies to reach the objective.  Some suggestions for further improvement are given below:

  • Line 60 ‘’Thus,…….. adsorbent’’: Please add the appropriate reference (https://doi.org/10.1016/j.catena.2019.104106)
  • In materials and methods, XAS analysis of Fe mineralogy is described in detail but I cannot find any data in the results and discussion section

Author Response

Authors’ response to comments

Thank you for your comments. We improved the MS according to your suggestion.

Reviewer 1

A well written paper on an important topic. The study contains valuable data, satisfactory selection of methodologies and strategies to reach the objective.  Some suggestions for further improvement are given below:

Comments: Line 60 ‘’Thus,…….. adsorbent’’: Please add the appropriate reference (https://doi.org/10.1016/j.catena.2019.104106)

Authors’ response: Thank you for the suggestion. The suggested reference was added.

Gasparatos, D. Massas, I., Godelitsas, A. 2019. Fe-Mn concretions and nodules formation in redoximorphic soils and their role on soil phosphorus dynamics: Current knowledge and gaps. Catena, 182, November 2019, 104106

Comments: In materials and methods, XAS analysis of Fe mineralogy is described in detail but I cannot find any data in the results and discussion section

Authors’ response: In section 3.1, the data is presented in Fig. 2. It is the results of LC fitting of XAS spectra. In the figure caption, we presented the result of fit. We edited the section to clarify the results of the LC fitting of reference compound analysis (See below). We also edited the figure caption of Figure 2B to clearly report the results of the LC fitting analysis.

“XRD analysis indicates that the mineralogy of iron oxides in the hybrid resin is predominantly goethite (Fig. 2A). The LC fitting of XAS reference spectra analysis also shows the same results (Fig. 2B). Oscillations and spectra features are nearly identical to those of goethite (Fig. 2B).  Ferrihydrite was not detected in the XAS analysis.”

Reviewer 2 Report

This paper provides important information about P desorption methods available in the field. The use of hybrid anion exchange resins as a P sink was very efficient in removing desorbed P from the agricultural soils. The iron oxide-coated hybrid anion exchange resins used in this study were promising, and were characterized using XRD and XAS to obtain the information about the mineralogy. The methodological approach seems solid and appropriate. Phosphate adsorption data were analyzed using Langmuir and Freundlich isotherm models. Furthermore, phosphate desorption kinetic were investigated in detail by pseudo-first-order model, pseudo-second-order model, Elovich model and Intra-particle diffusion model. The paper was well organized and the results are important for soil systems and environmental protection. There is no problem regarding English. Therefore, I feel this paper should be acceptable after some revision in view of the following specific comments.

(1) It would be better if the results (i.e., phosphate desorption kinetic by the hybrid anion exchange resins) are compared with those in other previous studies such as using other materials (not only compared to that of pure resin).

(2) It would be great for the authors to mention the adsorption and desorption mechanism of phosphate by the hybrid anion exchange resins in appropriate section. 

(3) It would be interesting for the authors could include the surface analyses of the hybrid anion exchange resins such as SEM-EDS or XPS etc. before and after desorption of phosphate.

Author Response

Authors' response to the comments:

Reviewer 2

This paper provides important information about P desorption methods available in the field. The use of hybrid anion exchange resins as a P sink was very efficient in removing desorbed P from the agricultural soils. The iron oxide-coated hybrid anion exchange resins used in this study were promising, and were characterized using XRD and XAS to obtain the information about the mineralogy. The methodological approach seems solid and appropriate. Phosphate adsorption data were analyzed using Langmuir and Freundlich isotherm models. Furthermore, phosphate desorption kinetic were investigated in detail by pseudo-first-order model, pseudo-second-order model, Elovich model and Intra-particle diffusion model. The paper was well organized and the results are important for soil systems and environmental protection. There is no problem regarding English. Therefore, I feel this paper should be acceptable after some revision in view of the following specific comments.

Authors’ response: Thank you for your time to review our MS.

Comments 1: It would be better if the results (i.e., phosphate desorption kinetic by the hybrid anion exchange resins) are compared with those in other previous studies such as using other materials (not only compared to that of pure resin).

Authors’ response: We appreciate the comments. We looked for previous studies about the P desorption and desorption kinetics using other materials in a mesh bag. We were not able to find in web of sci and google scholar. Our desorption method seems new to the field of “Soil Science”. It is our guess that other materials are too fine to trap in a mesh bag to conduct the desorption. They are difficult to separate from soil particles to assess the desorption. We feel that the comparison with the respective anion exchange resin (i.e., resin with the same polymer matrix) is most appropriate in this case.

However, we could add a few lines about a comparison of P adsorption capacity with other adsorbents (e.g., iron oxyhydroxides, ZVI, LDH, Zr- and La-containing hydroxides). Obviously, these materials cannot be used in this mesh bag desorption method because of small particle size. These materials also leak out from the mesh bags. Their small particle size is not suited for the proposed desorption method. We tested granule ZVI. When they in contact with water, ferrihydrite started to form via oxidative precipitation, and then ferrihydrite leaked out from a mesh bag. These adsorbents cannot be separated from soil particles to assess the P desorption process.

We just published a review paper in Advanced in Agronomy, vol 164. We talked about the P adsorption capacity of these adsorbents. We added the following paragraph in section 3.2 of the revised MS with a new reference.

Li, Z. and Arai, Y. 2020. Comprehensive evaluation of mineral adsorbents for phosphate removal in agricultural water. Advances in Agronomy. Volume 164. (In press)

“The P adsorption capacity of the hybrid resin at near neutral pH is within the same order of magnitude with that of other adsorbents (e.g., iron oxyhydroxides, zerovalent iron, layered double hydroxides, Zr- and La-containing hydroxides) (Li and Arai, 2020). However, the hybrid resin is more suitable for the proposed mesh bag-based desorption method than the other adsorbents because its large particle size makes easy to trap in a mesh bag and to recovery/separate from soil solutions.”

Comments 2: It would be great for the authors to mention the adsorption and desorption mechanism of phosphate by the hybrid anion exchange resins in appropriate section. 

Authors’ response:

About the adsorption mechanism, we have already reference papers about the inner-sphere adsorption of P in iron oxides in section 3.3.

“………………….which should be accounted for by the inner-sphere complexation of phosphate on the iron oxide coating in the hybrid resins (Arai and Sparks, 2001; Hongshao and Stanforth, 2001).”

About the desorption mechanism of P in the hybrid resin: P desorption process from the hybrid resin is via ligand exchange. Unless phosphate is desorbed with anions with greater shared charge, it cannot be desorbed P from the hybrid reason surfaces. We feel that the best place to add the P desorption mechanisms in the hybrid resin is the section 2.9. We justified the use of hydroxy anions as NaOH in an extracting solution.

“…To desorb chemi-sorbed P in the Fe oxide coating of the hybrid resin, anions with higher shared charge (e.g., hydroxyl ions) were necessary to extract adsorbed phosphate.

Comments 3: It would be interesting for the authors could include the surface analyses of the hybrid anion exchange resins such as SEM-EDS or XPS etc. before and after desorption of phosphate.

Authors’ response: It was our original plan to conduct SEM analysis but electron microscope labs at our institution have been closed due to Covid-19. We cannot do the SEM analysis now. They are currently disinfecting facilities but it is not clear when they will accept outside users. This includes XPS and other laser-based instrumentation. This is why this paper is heavy on macroscopic data. Thank you for understanding.

Round 2

Reviewer 2 Report

This is an interesting study denoting novel application of hybrid anion exchange resin for phosphate desorption kinetics in soils.

Based on the reviewers' comments, the manuscript is properly and sincerely corrected on the whole. Therefore, I feel this paper should be acceptable after minor revision in view of the following comments.

I recommend all the physical quantity such as k, q, C, t etc. in Table 5, Figure 6 and Figure 8 should be described italic.

Author Response

Reviewers' comments: I recommend all the physical quantity such as k, q, C, t etc. in Table 5, Figure 6 and Figure 8 should be described italic.

Response to the reviewer’s comments:

We italicized the symbols  in Table 5, Figure 6 and Figure 8. Thank you